# Transmission Power Control in Wireless Sensor Networks Using Fuzzy Adaptive Data Rate

**DOI:** 10.3390/s22249963

**Published:** 2022-12-17

**Authors:** Chung-Wen Hung, Yi-Da Zhuang, Ching-Hung Lee, Chun-Chieh Wang, Hsi-Hsun Yang

**Affiliations:** 1Department of Electrical Engineering, National Yunlin University of Science and Technology, Yunlin 64002, Taiwan; 2Department of Electrical and Computer Engineering, National Yang Ming Chiao Tung University, Hsinchu City 30010, Taiwan; 3Department of Electrical Engineering, Chienkuo Technology University, Changhua City 500020, Taiwan; 4Department of Digital Media Design, National Yunlin University of Science & Technology, Douliou 64002, Taiwan

**Keywords:** Internet of Things (IoT), wireless sensor network (WSN), adaptive rate control, transmission power control, fuzzy controller

## Abstract

As the technology of Internet of Things (IoT) becomes popular, the number of sensor nodes also increases. The network coverage, extensibility, and reliability are also the key points of technical development. To address the challenge of environmental restriction and deployment cost, most sensor nodes are powered by batteries. Therefore, the low-power consumption becomes an important issue because of the finite value of battery capacity. In addition, significant interference occurs in the environment, thereby complicating reliable wireless communication. This study proposes a fuzzy-based adaptive data rate for the transmission power control in wireless sensor networks to balance the communication quality and power consumption. The error count and error interval perform the inputs of a fuzzy system and the corresponding fuzzy system output is guard that is utilized for limiting the upper bounds of data rate and transmission power. The long-term experimental results are introduced to demonstrate that the control algorithm can overcome environmental interference and obtain low-power performance. The sensor nodes have reliable communication under an ultra-low-power consumption. The experimental results show that the total power consumption of the proposed approach has been improved 73% compared with the system without executing the algorithm and also indicate the Packet Error Rate (PER) is close to 1%. Therefore, the proposed method is suitable for the battery supply IoT system.

## 1. Introduction

With the advancement of technology, the concepts of smart manufacturing and smart city are popularized in the world. The Internet of Things (IoT) has been applied globally. Several technologies combined with IoT will bring some benefits, such as personnel cost reduction, real-time intelligent surveillance, etc. [1,2]. The wireless sensor networks (WSN) could address the challenges of power and signal installation, additional deployment costs, and application field limitation. In WSN, the sensing node is usually powered by batteries; therefore, battery life and maintenance fee become a challenge for IoT applications. This study proposes Adaptive Data Rate (ADR) and Transmission Power Control (TPC) using fuzzy controllers to reduce the power consumption and extend the battery life.

The common frequencies of wireless transmission are divided into Sub-1 GHz, 2.4 GHz, and 5 GHz. The high frequency has a faster data rate; however, the 5 GHz signals would cause a higher attenuation than the others when it passes through obstacles. The corresponding coverage is also less than Sub-1 GHz and 2.4 GHz [3,4,5,6]. However, the Sub-1 GHz has wider coverage and lower power consumption owing to the corresponding narrow bandwidth and low frequency.

Moreover, when the power consumption of sensors is not high in WSN, the overall power consumption is mostly consumed in wireless communications. If radio function could be turned off properly, the node powered by batteries would work for several years [7]. The types of Median Access Control (MAC) include scheduling-based [8,9,10] and competition-based [11,12,13]. The competition-based ones can accommodate more nodes without maintaining time synchronization. Although the throughput of scheduling-based is not good enough compared with competition-based system, it has the features of low-power and no package conflict in air [14]. In addition, the literature [15,16] utilizes time-slot scheduling architecture to implement a Time Division Multiple Access (TDMA) system. Every sensor node is assigned a particular time-slot and it only needs to turn on to communicate with the bridge when its time-slot is pulled. In other words, each node can keep turning off during other times to save more power. Considering the low-power and system complexity, a time-slot scheduling of TDMA is adopted in this study. 

In WSN fields, the power consumption of sensor nodes has become a key technology. The battery powered nodes are very convenient to place anywhere [17,18,19]; however, battery life and maintaining frequency is a challenge for manufacturing because it is a finite capacity. It may achieve a good communication quality and long transmission distance by transmitting with maximum power [20]. Under the condition of ensuring communication quality, adjusting the transmission power via TPC can effectively reduce power consumption [21,22,23,24]. In addition to reducing the transmission power, ADR also affects the overall power consumption. When the data rate is faster, the transmission time is shortened and the overall power is reduced [25,26]. Packet Error Rate (PER), Received Signal Strength Indicator (RSSI), and signal-to-noise ratio (SNR) are often adopted as references to reflect the current communication quality. The literature [27] indicates that RSSI is an unreliable parameter to present communication quality. A high RSSI does not imply a low PER because interference would increase RSSI. In [28], experimental results indicate the performance of Packet Delivery Rate (PDR) in different data rates. Regarding the same PDR, the higher the data rate the higher SNR; it needs higher communication quality to maintain the same PDR when delivery rate (DR) increases. An energy efficiency (EE) method is proposed to estimate which combinations of transmission power and data rate have the best value of EE [29].

In previous research [30,31], we have proposed a transmission power and data rate control algorithm to balance power consumption and communication quality. These studies take RSSI as one of the feedback parameters for balancing power consumption and communication quality. However, RSSI is not a reliable parameter for evaluating communication quality because it may be interfered with by noise. The control algorithms also adopt PER as feedback to avoid excessive errors, but it only reacts after errors occur. Considering the effect of noise, this study employs SNR as a feedback parameter. SNR is calculated by signal strength and noise, and it ameliorates the disadvantages when using RSSI and PER as the feedback parameters. The transmission power and data rate can be adjusted in advance of the errors occurring, preventing the rapid PER rises. In addition, the algorithm proposed in this study adopts mixed control, altering the transmission power and data rate simultaneously to obtain lower power consumption and better response time.

Therefore, to extend battery life and maintain communication quality, an ADR and TPC mixed control algorithm using fuzzy systems is proposed in this study. The control algorithm can lower the power consumption while maintaining communication quality. The error count and error interval perform the inputs of a fuzzy system, and the corresponding fuzzy system output is Guard that is utilized for limiting the upper bounds of data rate and transmission power. The long-term experimental results are introduced to prove that the control algorithm can overcome environmental interference and achieve a low-power performance. The sensor nodes have reliable communication under an ultra-low-power consumption. If it is powered by batteries of 1200 mAh, the power consumption of the proposed approach is improved by 73%. The test results also indicate that the PER of all the nodes of the experimental group is approximately 1%. The contribution of this paper is to apply the Fuzzy Logic Control into WSN to decrease the impact of interference on environment and maintain the link quality with low-power consumption. Moreover, the proposed scheme has been implemented in real-world settings to verify its effectiveness and reliability.

The rest of this paper is as follows. Section 2 introduces the system implementation, including the system architecture and implementation of TDMA. The corresponding fuzzy control algorithm and analysis of power consumption and performance in WSNs is introduced in Section 3, and Section 4 introduces the experimental results and discussions. Finally, the conclusion is given.

## 2. System Implementation

### 2.1. System Architecture

The system architecture is illustrated in Figure 1, and all the wireless devices such as root and sensor nodes are constructed with Texas Instruments CC1310 [32]. The system architecture comprises a grid-powered root, several bridges, and multiple battery-powered sensor nodes. The network topology indicated in the middle of Figure 1 could be a tree or mesh topology, etc., and every bridge can connect with several sensor nodes. The connection between bridge and sensor nodes is a simple star topology for low-power and complex implementation concerns. This paper also proposes a TDMA protocol with data rate and transmission power control algorithm to reduce the power consumption of sensor nodes but maintain the communication quality between bridge and sensor nodes.

### 2.2. Implementation of TDMA

The TDMA scheme adopted in this paper is illustrated in Figure 2. Obviously, time-slots are divided into odd and even slots to be compatible with two topologies; the bridge communicates with sensor nodes for obtaining sensor data and power control in odd slots, and it updates topology information to sensor nodes in even slots. Sensor nodes would be assigned two slots (odd and even) when they have built a connection with the bridge. In the odd slot, the required packet is utilized for transmitting the control packet from bridge to sensor node. The control packet includes transmission power control, data rate control, and slot request, and it is also used for time synchronization. Response packet is utilized for response after sensor nodes receive the required packet from the bridges. The even slots are utilized for running low-power control algorithms, updating topology information, or have a handshaking to other nodes in network topology.

## 3. Adaptive Fuzzy Control Algorithm and Analysis of Power Consumption 

### 3.1. Power Consumption in Different Data Rate and Transmission Power

The transmission power range is 0 to 14 dBm in CC1310, and its modulation is 2-FSK (Frequency Shift Keying). The configurations of eight data rates used here are introduced in Table 1. It indicates the frequency deviation and RX BW (Received Bandwidth) of eight different data rates. These parameters of each data rate are configured according to [30] provided by Texas Instruments. Table 2 presents the actual current corresponding to each transmission power, and the current in this table is the average value. The values of average current corresponding to each transmission power were measured by ourselves, using a high precision current meter. To effectively evaluate and analyze the relationship between data rate and power consumption, the required time for transmission is also an important parameter for power consumption analysis. The transmission time can be simply calculated by
(1)$$Transmission\;time=\frac{Data\;length\;\left(bits\right)}{Data\;rate\;\left(bps\right)}$$
where the data length is 16 bytes.

The corresponding power consumption is calculated by
(2)Power consumption=Current×Transmission time

The relation of power consumption, data rate, and transmission power is illustrated in Figure 3. Obviously, using larger transmission power to communicate will consume more power. Furthermore, increasing the data rate, which means lowering the transmission time of fixed-length packet, will reduce opening time for radio frequency (RF) and help the node to enter the sleep mode sooner, which in turn decreases the power consumption.

### 3.2. Relationship between PER and SNR

Because RSSI is sensitive to environmental interference, it is an unreliable parameter; therefore, it will be difficult to evaluate the communication quality. Therefore, SNR is adopted for evaluating communication quality. According to [33], minimum detectable signal (MDS), also known as receive sensitivity, can be calculated by Formula (3). This paper takes the average SNR at PER 1% as receiver sensitivity.
(3)Sensitivity(dBm)=SNR+Noise

The demodulator of CC1310 is a non-coherent demodulator of 2 FSK. From [34] we can learn the method to calculate the relation between Bit Error Rate (BER) and SNR, and the result can be extended to the relation of PER and SNR by importing the relation between BER and PER. We can obtain
(4)$$PER=1-{\left(1-\frac{1}{2}e^{-\frac{E_{b}}{2N_{0}}}\right)}^{n}$$
where $\frac{E_{b}}{N_{0}}$ can be obtained by
(5)$$\frac{E_{b}}{N_{0}}=10^{\left(SNR+offset\right)/10}$$
where *n* denotes data length.

Each data rate has a different offset expressed in Equation (4). Figure 4 is illustrated to measure the relationship between PER and SNR. In this architecture, two CC1310 devices are utilized as the transmitter and receiver. The transmitter connects with a programmable attenuator. The programmable attenuator has an attenuation range from 0 to 120 dB and can be fine-tuned in 0.5 dB. After the transmitter had transmitted 1000 packets, the receiver recorded average SNR and PER for each attenuation. Owing to uncertain interferences around the environment, the experiment was made in an anechoic chamber.

The offsets of each data rate are presented in Table 3. Substitute the offsets into Equation (4) to obtain the receiver sensitivity under PER 1%. Table 4 presents the values of SNR for each data rate under PER 1%, and the fixed value indicated in Table 4 is substituted into Equation (3) to obtain a sensitivity. The sensitivity is a dynamic value determined by time-varying noise. When the RSSI which is obtained from the received packet is higher than sensitivity, it means that the PER will be below 1% in theory. The lower the sensitivity, the better the receiving ability is. When the noise is fixed, the higher data rate results in a higher sensitivity and worse receiving ability; however, it reduces the power consumption by reducing the transmission time. 

Figure 5 illustrates the relation curves of four different data rates. To save more power, it can be achieved by adjusting the data rate and transmission power. There is an example given below to explain the algorithm concept proposed in this study. The transmission power of a sensor node is 10 dBm, its data rate is 50 kbps, and its SNR locates at 5.22 dB. In these conditions, the power consumption can be reduced further without changing the communication quality by simultaneously altering data rate and transmission power. The original and possible combinations which comprise data rate and transmission power are presented in Table 5. The combination of most power saving is 200 kbps and 13 dBm, and it can save power approximately 78% off the original set. However, 300 kbps is not a suitable data rate for this condition because its transmission power requires 16 dBm; this is explained by the transmission power beyond the adjustment range to the device. In this case, the power consumption can be reduced by increasing data rate and transmission power simultaneously but not changing the communication quality.

### 3.3. Fuzzy Control Algorithm Design

Fuzzy logic system is a method of reasoning that resembles human reasoning. The approach of FL imitates the process of decision making in humans that involves all intermediate possibilities between binary logic. Several fuzzy control approaches have been applied to the industry with several important theoretical and successful results [33,34,35,36,37,38]. The control approach based on human experience is acting in fuzzy controls by expressing the control requirements and elaborating the control signal in terms of IF–THEN rules in which *i*th rule can be represented as
Rule *i*: IF (antecedent) THEN (consequent)(6)
where the antecedent (premise) refers to the found-out situation, and the consequent (conclusion) refers to the measures which should be made on the decision.

Mamdani [39] and Sugeno [40] are two well established types of Fuzzy Inference Systems (FIS). In this study, the Mamdani fuzzy inference is adopted to create a control system for adaptive data rate. In a Mamdani system, the output of each rule is a fuzzy set which means it has advantages of expressive power and interpretable rule consequent, while Sugeno does not [41]. Therefore, Mamdani FIS is adopted in this paper. Although Segeno has better effectiveness than Mamdani, the proposed system decreases the computational complexity by using a look-up table to obtain the final output value from a fuzzy controller.

Because multiple factors affect wireless transmission, the relation of PER and SNR does not always fit to theory. This study proposes a low-power control algorithm based on Mamdani fuzzy controller to solve time-varying and nonlinear problems. The system can achieve a more precise corresponding result and better control effect with larger range of setting the MF or more designing rules. However, the proposed system needs to implement the fuzzy controller into the target device. Considering the storage space of the device, we chose suitable parameters of membership function by observing the long-time testing data collected in the past. The following five linguistic values are used, VL (Very Low), L(Low), M(Medium), H(High), and VH (Very High). To conquer time-varying interference, this study uses error count and error interval as the inputs of a fuzzy control system. The corresponding membership functions are illustrated in Figure 6. Error count is the number of errors in 128 moving windows, which can also be expressed as PER; when it has 1 error in 128-windows, PER would be 0.78% and it becomes 1.56% after receiving another error in 128-windows which exceeds the control objective of the system. As a result, there are more membership function overlaps at the range of 1 to 3, the place which is more sensitive and needs a precise control, when designing the membership function of error count. Error interval is calculated by the interval away from the latest error. These input parameters can represent communication quality and stability. The membership functions for output are illustrated in Figure 7. The output value, guard, is adopted for limiting the upper bounds of data rate and transmission power. The corresponding twenty-five fuzzy rules are presented in Table 6. It keeps RSSI away from sensitivity for getting a promotion in communication quality when errors are close to H or VH and error interval is close to L or VL. On the contrary, it attempts to be close to sensitivity for decreasing the power consumption when errors are close to L or VL and error interval is close to H or VH. The relations of inputs and output according to the center of gravity (COG) method for defuzzification are illustrated in Figure 8. To reduce the real-time computational effort, the fuzzy surface illustrated in Figure 8 is transferred to a look up table for on-line implementation. The look up table is organized by an error count that ranges from 0 to 11 and an error interval ranging from 0 to 127, and its size is 1536 bytes.

According to the analysis of transmission performance, power consumption, and fuzzy controller, a fuzzy-based data rate and transmission power control algorithm was proposed. The flow chart for the control algorithm is illustrated in Figure 9. After receiving the response packet, it performs statistics and analysis about the packet. Furthermore, all the input parameters are substituted into the fuzzy controller to obtain a crisp value for determining data rate and transmission power. If there is any interferece during the transmission time, it will be reflected in the value of SNR and PER. As a result, these two parameters are used to adjust the input of the fuzzy system. We calculate the differences in RSSI and sensitivities of eight data rates as
(7)$$RSSI-Sensitivity_{i}\geq Guard,\;where\;i=0,\;1,\;\ldots ,\;7$$

The algorithm chooses the highest data rate that satisfies the condition via Equation (7); and it adopts $TP=TP_{Max}-Difference$ to calculate the transmission power; where $TP_{Max}$ represents the highest tramsmission power of the device. Finally, the data rate and transmission power are updated to the configuration table.

## 4. Experimental Results and Discussions

### 4.1. Allocation of Bridge and Sensor Nodes

The TDMA polling period is 10 s in the experiments. The root, bridge, and 10 sensor nodes are placed in different positions. The sensor nodes are divided into experimental and control groups, and placed in five different locations for testing. The experimental site is located in the Sixth Hall of Engineering, National Yunlin University of Science and Technology, Taiwan. Figure 10 illustrates the configuration diagram of the nodes’ position. The red dots are the root and bridge, and the green dots are the sensor nodes. The sensor nodes are set in five different positions, and there are two sensor nodes for the experimental and control groups in each position. Each sensor node is numbered 1 to 10 for analyzing experimental results easily; odd numbers 1, 3, 5, 7, 9 are experimental groups, and even numbers 2, 4, 6, 8, 10 are control groups.

Table 7 presents the data rate and transmission power used by each sensor node in the long-term test. The parameter combination of each control group is the most commonly used combination under PER 1% after being tested by the algorithm group for a period of time.

### 4.2. Experimental Results

The experimental results are presented in Table 8. In addition to the experimental data, the table briefly describes the interferences at each experimental location. The long-term test results indicate that the overall PER statistical results of the experimental group all converged to approximately 1% of the control target, whereas the control group with a fixed data rate and transmission power has different PER performance owing to environmental interference. In general, the sensor nodes in other positions are relatively power-saving in terms of the overall average current, except for sensor nodes 5 and 6 that have less power-saving effect; their power consumption is quite similar. When the experimental group is in good communication quality, it will reduce power consumption as much as possible so that RSSI is close to sensitivity; however, when the experimental group is in the bad communication quality, it will increase power consumption appropriately to keep RSSI away from sensitivity to obtain better communication quality.

Figure 11 illustrates the long-term test results of sensor nodes 1 and 2 in the experimental and control groups, respectively. This position is the farthest from the bridge. The PER of the experimental group is 1.04%, and the location is interfered with by people walking around. However, the data rate and transmission power are appropriately adjusted by the algorithm so that the RSSI changes are similar in the daytime and at night. The PER of the control group is approximately 0.07%, and is located at the entrance of the first floor; therefore, there are several people walking around and this will cause interference, which leads to obvious differences in RSSI between the daytime and night. The RSSI in the daytime changes drastically; however, the RSSI at night has negligible change because there are fewer people walking around at night.

Regarding the current consumption, the average transmission and overall average current of the experimental group is approximately 0.71 and 19.49 µA, respectively, whereas the average transmission and overall average current of the control group is approximately 1.12 and 20.57 µA, respectively. The experimental group saves the average transmission current and is approximately 36.01%, whereas the overall average current is approximately 5.26%.

Figure 12 illustrates the moving average PER curve for a sliding window of 1000 of sensor nodes 5 and 6. The PER of the experimental group with the algorithm is approximately 1%, whereas the PER value of the control group is less than 1% most of the time. The control group is located outside the office on the second floor; therefore, there are more people walking around to cause interference. From the statistical results, the overall PER of the control group is low, but there are several PER values that are high owing to external interference. Compared to the PER curve of the experimental group at the same timeline, the PER is lower than the control group. Although the power consumption of the two are similar, the PER performance of the experimental group is relatively stable.

All sensor nodes are powered by two AAA batteries in series, and the power that can be supplied is approximately 1200 mAh. Herein, a simple straightforward formula is adopted to evaluate the battery life in years as
(8)$$Battery\;Life\left(Years\right)=\frac{Battery\;Capacity\left(mAh\right)}{365\times 24\times I_{avg}}$$
where the battery capacity in this formula is 1200 mAh and $I_{avg}$ denotes the overall average current of the sensor node in mA.

Table 9 presents the battery life table of each sensor node from the experimental group calculated by the above method. The table also includes the estimated results of the most power consumed parameters. The calculation results ignore the leakage of the battery itself. The battery life with the transmission parameters of the most power consumed is approximately 2.04 years. However, the best battery life is approximately 7.26 years, and the shortest battery life is approximately 6.82 years for the sensor nodes with the proposed fuzzy control algorithm. It can be observed from the calculation results that the sensor node can operate for more than two years with TDMA architecture. Furthermore, with the control algorithm of ultra-low-power consumption, it can extend the battery life to more than 6.8 years. The above proves that the control algorithm proposed in this study has an excellent effect on power saving.

## 5. Conclusions

Based on the Sub-1 GHz, this paper proposed a fuzzy-based adaptive data rate and transmission power control algorithm, then demonstrated that the sensor node has the characteristic of extremely low power consumption. As for the topology, it retained the capability of network topology and low power consumption of star topology. The actual field test was affected by several external factors. The proposed control algorithm adaptively controlled the data rate and transmission power to overcome the effect from external environment and noise interference. The long-term experimental results demonstrated that the PER of the sensor node from the experimental group is mostly controlled at approximately 1% to maintain a balance between communication quality and power consumption. To retain the features of low power consumption and communication quality, the algorithm tried to reduce the transmission power to obtain lower power consumption when the communication reached a level of stability. The proposed algorithm effectively avoided the high PER from the sudden appearance of communication errors and maintained a level of communication quality. The experimental results of power consumption indicated that the battery life of each experimental group node is more than 6.8 years (improvement by 73% compared with previous results). However, the battery life will only be about 2 years in order to keep the PER below 1% with the most power consumed combination. The control algorithm proposed in this paper has the characteristics of maintaining stable communication and power saving, although the battery life of 6G WSN is expected to be 20 years, which is slightly larger than the result of this paper. Nevertheless, the experimental data are obtained from the long-time testing setting in the realistic field. Its feasibility and effect are verified. To summarize, the proposed method with above advantages can provide a better solution for WSN based on current techniques. For future work, the technique of Adaptive Modulation and Coding (AMC) and Channel Estimation (CE) can be implemented in this system to achieve a better communication quality with lower power consumption.

## Figures and Tables

**Figure 1 sensors-22-09963-f001:**
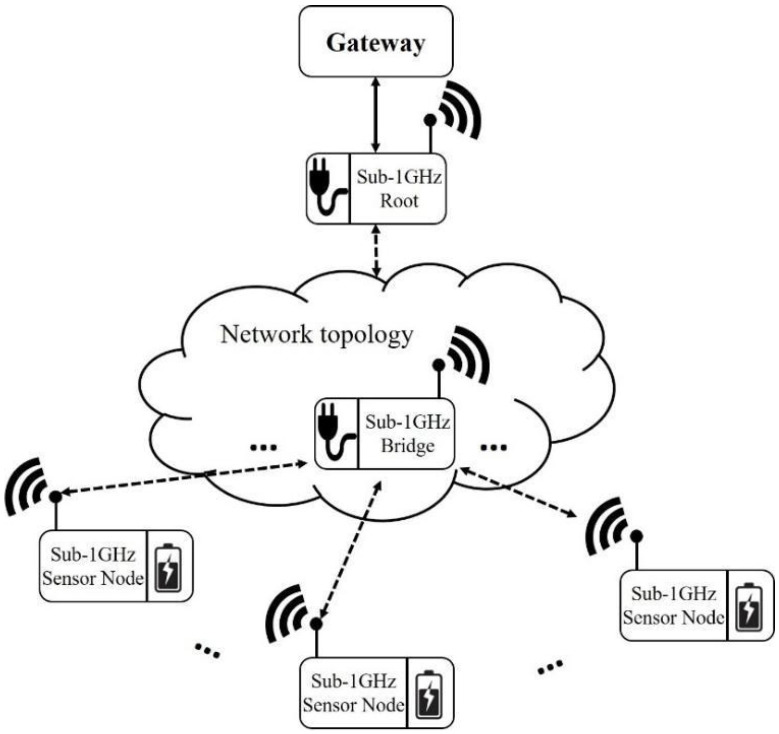
System architecture.

**Figure 2 sensors-22-09963-f002:**
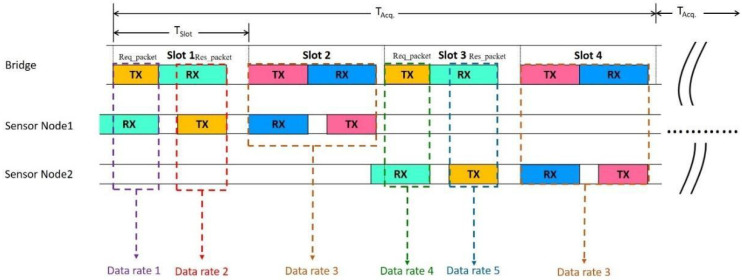
Schematic diagram of TDMA.

**Figure 3 sensors-22-09963-f003:**
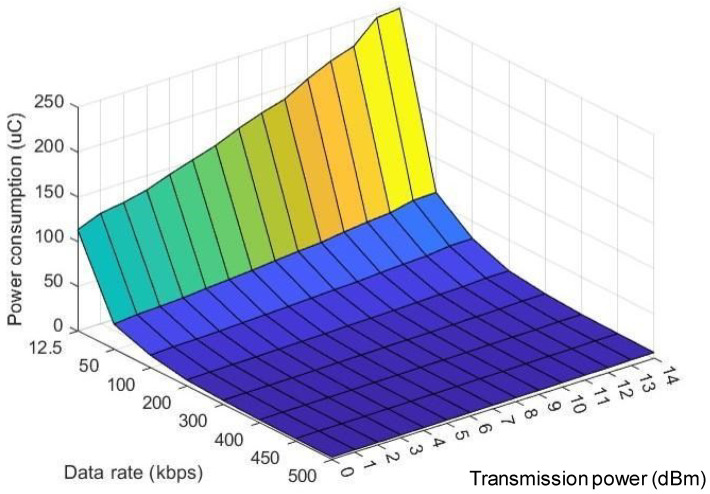
Relationship between power consumption, data rate, and transmission power.

**Figure 4 sensors-22-09963-f004:**
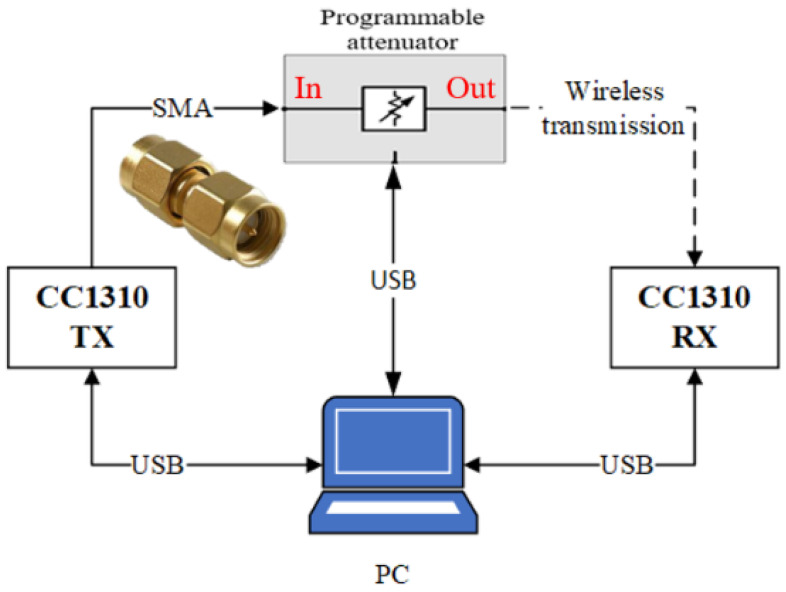
Experimental architecture for relation of PER and SNR.

**Figure 5 sensors-22-09963-f005:**
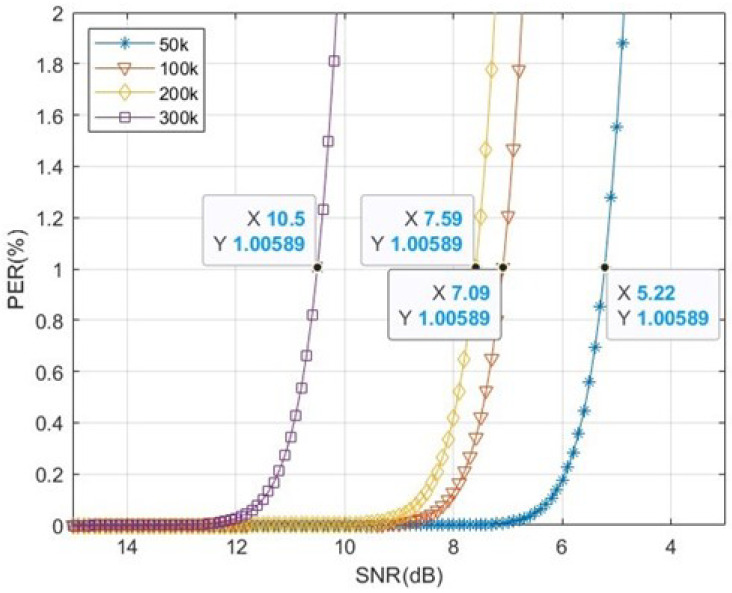
Relationship between PER and SNR in different data rates.

**Figure 6 sensors-22-09963-f006:**
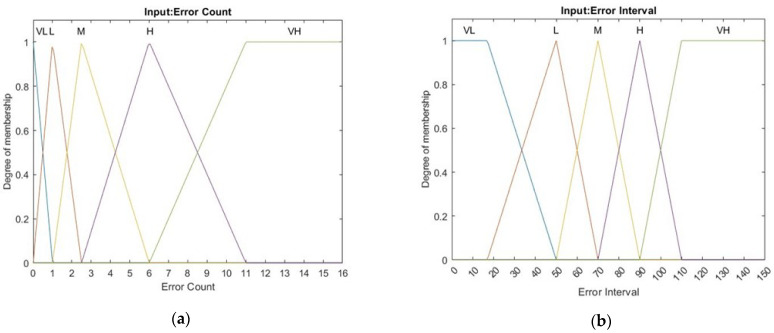
Membership functions for antecedent parts: (**a**) error count and (**b**) error interval.

**Figure 7 sensors-22-09963-f007:**
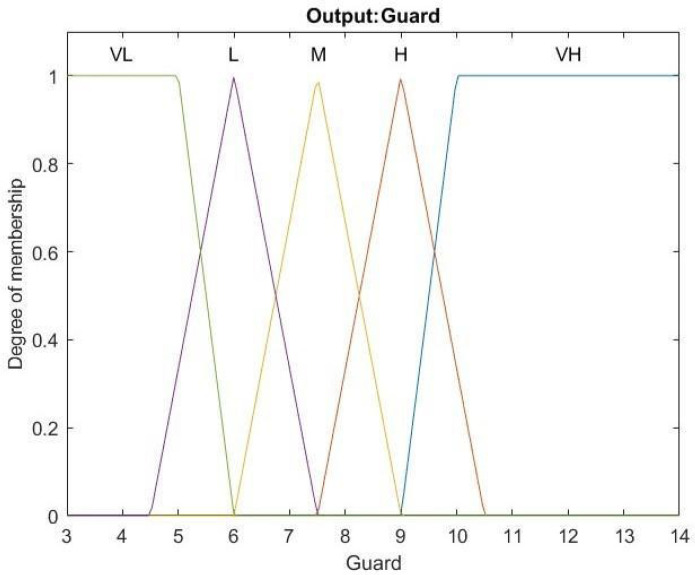
Membership functions for Guard.

**Figure 8 sensors-22-09963-f008:**
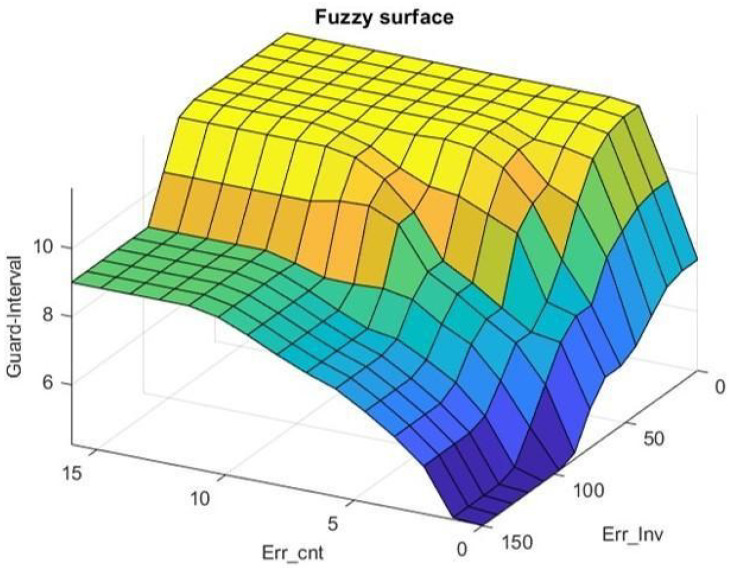
Relations of error count, error interval, and guard.

**Figure 9 sensors-22-09963-f009:**
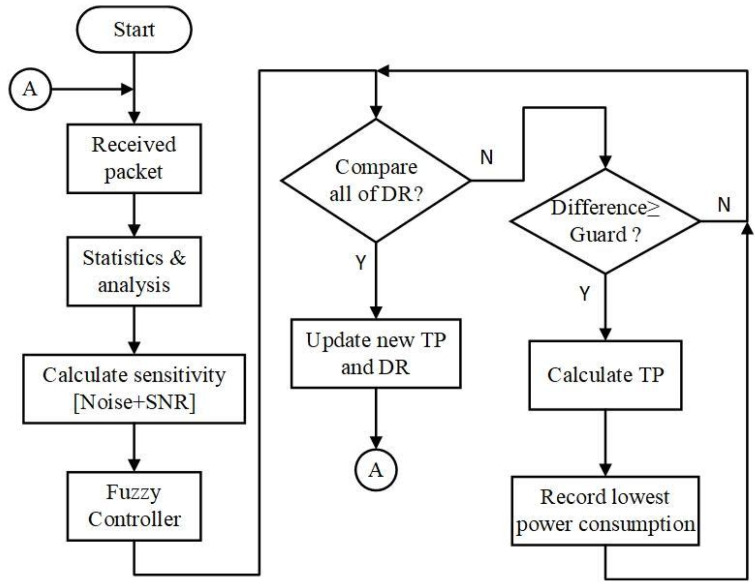
Flow chart for the control algorithm.

**Figure 10 sensors-22-09963-f010:**
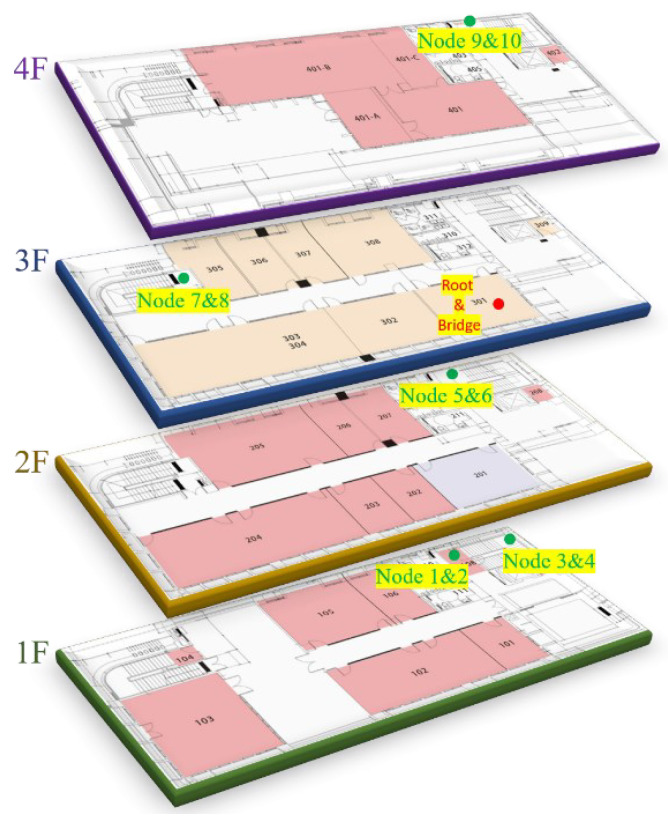
Configuration diagram of the nodes’ positions.

**Figure 11 sensors-22-09963-f011:**
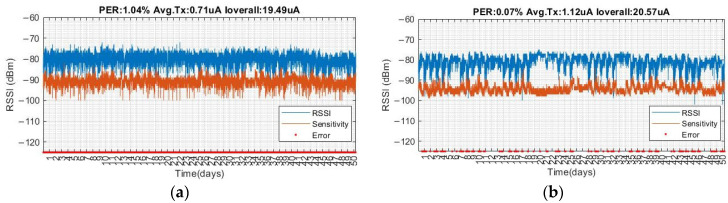
Experimental results for (**a**) Node 1 and (**b**) Node 2.

**Figure 12 sensors-22-09963-f012:**
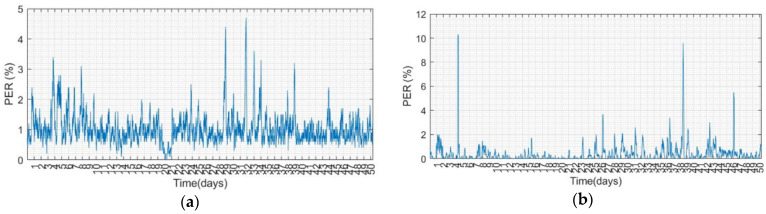
PER in 1000-window of moving average for (**a**) Node 5 and (**b**) Node 6.

**Table 1 sensors-22-09963-t001:** Parameters of each data rate.

Data Rate	Deviation	RX BW	Data Rate	Deviation	RX BW
12.5 kbps	5 kHz	49 kHz	300 kbps	105 kHz	622 kHz
50 kbps	25 kHz	98 kHz	400 kbps	140 kHz	622 kHz
100 kbps	45 kHz	196 kHz	450 kbps	155 kHz	784 kHz
200 kbps	70 kHz	311 kHz	500 kbps	175 kHz	1243 kHz

**Table 2 sensors-22-09963-t002:** Average current.

Transmission Power	Average Current	Transmission Power	Average Current	Transmission Power	Average Current
14 dBm	24.138 mA	9 dBm	18.181 mA	4 dBm	13.992 mA
13 dBm	23.894 mA	8 dBm	17.460 mA	3 dBm	13.009 mA
12 dBm	21.575 mA	7 dBm	16.569 mA	2 dBm	12.447 mA
11 dBm	20.749 mA	6 dBm	15.496 mA	1 dBm	12.064 mA

**Table 3 sensors-22-09963-t003:** Offsets of each data rate.

Data Rate (kbps)	12.5	50	100	200	300	400	450	500
Offset	9.73	7.27	5.39	4.89	1.99	1.04	−0.44	−3.64

**Table 4 sensors-22-09963-t004:** SNR of each data rate under PER 1%.

Data Rate (kbps)	12.5	50	100	200	300	400	450	500
$SNR_{PER\;1\%}$ (dB)	2.75	5.22	7.10	7.60	10.50	11.45	12.93	16.13

**Table 5 sensors-22-09963-t005:** Comparison table in different parameters.

	Data Rate 50 kbps	Data RATE 100 kbps	Data Rate 200 kbps
PER	~1%	~1%	~1%
SNR	5.22 dB	7.09 dB	7.59 dB
Transmission Power	10 dBm	12 dBm	13 dBm
Power Consumption	35.64 uC	15.93 uC	7.72 uC
Saving Rate	╳	55.30%	78.34%

**Table 6 sensors-22-09963-t006:** Fuzzy rules table.

	Error Count	VL	L	M	H	VH
Error Interval						
VL		M	H	VH	VH	VH
L		L	M	H	VH	VH
M		L	M	M	H	VH
H		VL	L	M	H	H
VH		VL	VL	L	M	H

**Table 7 sensors-22-09963-t007:** Data rate and transmission power of sensor nodes.

Node No.	Data Rate	Transmission Power
1, 3, 5, 7, 9	Adjust according to the algorithm	
2	8 dBm	200 kbps
4	9 dBm	100 kbps
6	8 dBm	400 kbps
8	5 dBm	450 kbps
10	10 dBm	400 kbps

**Table 8 sensors-22-09963-t008:** Statistics of each sensor node.

Node No.	PER	Average Transmission Current (µA)	Overall Average Current (µA)	Briefly Describe the Effects of Interference
1	1.04%	0.71	19.49	It is at the entrance of the first floor and farthest from the bridge. It is more obviously affected by people walking around.
2	0.07%	1.12	20.57	
3	0.77%	1.02	20.09	It is in the stairwell and people walk around. The elevator also starts and stops, causing interference.
4	1.51%	2.33	23.30	
5	1.03%	0.66	19.46	It is outside the office and is more obviously affected by people walking around during the day.
6	0.40%	0.56	19.25	
7	0.42%	0.30	18.86	It is on the same floor as the bridge and few people will pass by.
8	1.52%	0.42	19.02	
9	1.22%	0.59	19.24	It is outside the classroom and there are some people who walk around occasionally.
10	0.46%	0.63	19.31	

**Table 9 sensors-22-09963-t009:** Battery life of sensor nodes.

Node Number	Battery Life
1	7.03 years
3	6.82 years
5	7.04 years
7	7.26 years
9	7.12 years
Parameters which consume the most power	2.04 years

## Data Availability

Not applicable.

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
