# Peer review of "Transmission Power Control in Wireless Sensor Networks Using Fuzzy Adaptive Data Rate"

_sensors, 2022, doi:10.3390/s22249963_

Round 1
Reviewer 1 Report
This paper proposes a fuzzy‐based adaptive data rate for transmission power control in wireless sensor networks to balance communication quality and power consumption. The error count and error interval perform the inputs of a fuzzy system, and the corresponding fuzzy system output is a guard that is used to limit
The authors have put great effort into this paper.
Please improve the quality of figure 1.
Well done!
Author Response
We would like to thank you for your insightful comments and valuable suggestions. This paper has been carefully revised according to their constructive comments/suggestions. In addition, the manuscript has been revised by Editing Office. Please see the revised version and reply form.

Reviewer 2 Report
In order to balance communication quality and power consumption in wireless sensor networks, this research suggests a fuzzy-based adaptive data rate. During the revision, the following factors need to be taken into account.
The language needs to be refined.
Most of the cited reference works are conference papers. It is recommended to use standard journals for citation and the corresponding works should be highlighted in literature.
When using abbreviations for the first time, they should be defined. For example, PER is not defined in the abstract. Later it is defined in the Introduction section.
In abstract, 73% improvement in power consumption is mentioned. This is computed with respect to which scheme(s)?
The interference factor is mentioned in the abstract. But it is not emphasized in the work.
The research gaps and major contributions must be listed at the end of Introduction section.
Figure 1 is missing. Two times Figure 2 is mentioned in the captions.
Mention the references for Tables 1 and 2.
Add an explanation for the surface plot in Figure 3.
Validate equation (3), because SNR itself includes noise. Again another noise is added to SNR. If it is referred from some papers, give proper citation.
Justify the reason to choose Mamdani fuzzy system over using other fuzzy systems like Sugeno.
The range used to construct membership plot for error count is nonuniform. How it is decided?
The number of membership functions, their range, the number of rules-how these are decided?
"The long‐term experimental results are introduced to show that the control algorithm can overcome environmental interference and get a low‐power performance. The sensor nodes have reliable communication under an ultra‐low‐power consumption. If it is powered by batteries of 1200mAh, the power consumption of the proposed approach has been improved 73%. The test results also indicate the PER of all the nodes of the experimental group are close to 1%"-The above lines mentioned in Introduction is reused from abstract. At least paraphrase them.
The results lack insights and are not presented properly.
In 6G, batteries used in WSN are expected to have a life time of 20 years. As per your results, it is mentioned around 7 years. How your work will address this gap? Highlight the short comings and future works.
The quality of the figures should be improved.
Author Response

(The authors gave the same response as above.)

Reviewer 3 Report
There are many phrases difficult to understand, they are highlighted in the attached file. The work itself seems sound.

Author Response

(The authors gave the same response as above.)

Round 2
Reviewer 2 Report
All my comments were addressed by the authors.